# Required Spine Optional Limbs: Heterogeneous Federated Learning via Backbone-sharing and Activation-guided Selection

Mingsheng Cao [1]  Hongliang Chen [1]  Ming Hu [2]  Fei Gao [3]  Qiaolong Ding [1]  Wenke Huang [4]  Xiaofei Xie [5]
Junlong Zhou [6]

## Abstract

Although Federated Learning (FL) enables privacy-preserving cross-device collaboration, its practical deployment is hindered by heterogeneous data and hardware resources. Sub-model extraction has therefore emerged as a common strategy for collaborative training across heterogeneous devices. However, existing sub-model extraction methods rely on stochastic or rule-based neuron selection, leading to parameter conflicts under non-IID data and insufficient model optimization across heterogeneous devices. To address this problem, this paper presents a novel sub-model extraction-based FL framework, named SpineFL, which adopts a backbone-sharing mechanism and an activation-guided pruning strategy for sub-model extraction. Specifically, SpineFL decomposes each global model layer into two portions: i) a mandatory backbone shared by all the sub-models to maintain model generalization, and ii) a dynamic portion for sub-model extraction. SpineFL adopts the activation-guided selection strategy to probabilistically select neurons according to their activation frequency from the dynamic portion to generate a sub-model, where neurons exhibiting higher historical activation are more likely to be included, thereby simultaneously addressing parameter conflicts while preserving selection diversity. Experimental results demon-
strate that compared with state-of-the-art heterogeneous FL methods, SpineFL can achieve up to 3.28% accuracy improvement. The source code is available at https://github.com/alogrom/SpineFL.

## 1. Introduction

Federated Learning (FL) (McMahan et al., 2017; Yang et al., 2019; Wang et al., 2024b; Huang et al., 2024; Hu et al., 2024b; Qi et al., 2025; Huang et al., 2026; Li et al., 2025b) has gained widespread attention as a privacy-preserving decentralized learning paradigm that enables massive devices to collaboratively train a deep learning model without exposing any privacy-sensitive data. The growing adoption of edge computing and advanced mobile chips has positioned FL as a promising enabler for Artificial Intelligence of Things (AIoT) systems (Zhang et al., 2021; Wu et al., 2023; Hu et al., 2023; Li et al., 2025a). Typically, in AIoT systems, FL involves a central server and multiple local devices. In each FL training round, the server dispatches the global model to activated devices as local models for local training and aggregates all the trained local models to update the global model. However, due to AIoT systems typically involving massive heterogeneous devices, the performance of FL is seriously limited by to the constraints of memory, computation, and communication resources on devices (Cui et al., 2021; Liu et al., 2021; 2022). Specifically, due to the heterogeneity among devices, a critical bottleneck lies in selecting an appropriate global model size that simultaneously satisfies the resource constraints of all participating devices while maintaining training effectiveness. Typically, large models have strong reasoning capabilities, but since they cannot be trained on most devices, it is difficult for them to be adequately trained with sufficient data. In contrast, while small models can be fully trained, their reasoning abilities are limited.

To improve the performance of FL with heterogeneous devices, existing methods attempt to use heterogeneous models for FL training rather than the same global model, which can be classified into two categories, i.e., completely heterogeneous methods and sub-model extraction-based meth-

---

*Equal contribution [1] Sichuan Provincial Key Laboratory of Network and Data Security, University of Electronic Science and Technology of China, Chengdu, China [2] Software Engineering Institute, East China Normal University, Shanghai, China [3] College of Computer Science and Technology, National University of Defense Technology, Changsha, China [4] College of Computing and Data Science, Nanyang Technological University, Singapore [5] School of Computing and Information Systems, Singapore Management University, Singapore [6] School of Computer Science and Engineering, Nanjing University of Science and Technology, Nanjing, China. Correspondence to: Ming Hu <mhu@sei.ecnu.edu.com>.

ods. Typically, completely heterogeneous methods (Li & Wang, 2019) assign models with completely different model structures to each device and adopt knowledge distillation technologies (Wang et al., 2023; Li & Wang, 2019) or prototypes (Tan et al., 2022; Fu et al., 2025; Qi et al., 2023) to achieve knowledge sharing among heterogeneous devices. Due to depending on additional public datasets, the application of completely heterogeneous methods is seriously limited. Sub-model extraction-based methods (Diao et al., 2020; Ilhan et al., 2023; Alam et al., 2022; Wang et al., 2024a; Liao et al., 2024; Chen et al., 2024; Xu et al., 2024; Wang et al., 2024c) generate a sub-model by pruning a global hypernetwork according to the hardware resources of a target device. Due to without the requirement of the additional public dataset, sub-model extraction has emerged as a widely-used technical solution to address the problem of device heterogeneity.

Although existing sub-model extraction methods enable collaborative training across heterogeneous devices, they still suffer from performance degradation due to data heterogeneity. Specifically, heterogeneous data results in different optimal directions of local models (Karimireddy et al., 2020; Hu et al., 2024a; Xia et al., 2025; Wu et al., 2026; Meng et al., 2024), resulting in serious parameter conflict problems. Typically, existing sub-model extraction methods adopt parameter sharing (Diao et al., 2020) or rotation parameter selection (Alam et al., 2022) strategies to generate sub-models. Specifically, the parameter sharing strategy guarantees that the small sub-model is nested within the large sub-model. The rotation parameter selection strategy selects some parameters of each layer in turn to generate sub-models. Due to a lack of a wise neuron selection strategy, the mainstream sub-model extraction methods encounter serious performance degradation in data-heterogeneous scenarios. The state-of-the-art methods, such as FedDSE (Wang et al., 2024a), attempt to select specific neurons for a specific device according to its data distribution. However, such a strategy causes a lack of diversity in neuron selection, as many neurons end up being trained by only a small subset of devices. This strategy leads to the model preferring to learn features of specific device data, resulting in poor generalization. Therefore, *how to wisely extract sub-models to achieve high performance FL training in both data and devices heterogeneous scenarios is a significant challenge in FL.*

Intuitively, to ensure the generalization of the global model while alleviating the performance degradation from gradient conflicts caused by data heterogeneity, all the sub-model can share a part of the parameters as the backbone to facilitate the global model to learn more generalized features. To alleviate the gradient conflicts, the sub-model extraction strategy can select the neurons in the unshared part according to the data distribution of each client. In addition,

according to the observations in (Li et al., 2017; Jia et al., 2024), typically pruning parameters in deep layers can cause less performance degradation than pruning those of shallow layers. Based on the above observations, to improve the FL performance, the sub-model extraction strategy can select more neurons in shallow layers and prefer to prune neurons in deep layers to generate sub-models.

Inspired by the above motivation, to improve the training performance in both data and devices heterogeneous scenarios, this paper proposes a novel sub-model extraction-based framework named SpineFL, which adopts a backbone sharing mechanism and an activation-guided neuron selection strategy for sub-model extraction. Specifically, to ensure the generalization of the global model, SpineFL decomposes the global model into two parts, the global backbone and the dynamic portion, based on the hardware resources of the target device. SpineFL selects neurons from the dynamic portion together with the full backbone portion to extract sub-models. Note that sub-models share the same backbone architecture across sizes, with smaller backbones being subsets of larger ones. To alleviate the performance degradation caused by heterogeneous data, SpineFL integrates the activation-guided neuron selection mechanism to wisely select neurons in the dynamic portion according to the hardware resources of the target device. In summary, this paper has the following three main contributions:

- We propose a sub-model extraction-based framework named SpineFL, which adopts a layer-wise backbone sharing strategy to decompose the global model into a global backbone portion as the bedrock for all the sub-models and a dynamic selection portion for heterogeneous part extraction.

- We design an activation-guided neuron selection mechanism to wisely extract sub-models based on the hardware resources of the target device.

- We conduct extensive experiments on well-known datasets to demonstrate the effectiveness of SpineFL.

## 2. Preliminaries and Related Work

### 2.1. Preliminaries of Federated Learning

Typically, Federated Learning (FL) consists of a cloud server and multiple local devices. The goal of FL is to train a global model $w_{glb}$ that maps the input space $X$ to the output space $Y$. Assume that there are $|Dev|$ devices and each device $i$ maintains a local dataset $D_i = \{d_{i,1}, d_{i,2}, ..., d_{i,n_i}\}$, where $d_{i,j} = \{x_{i,j}, y_{i,j}\} \in X \times Y$. In addition, assume that $f_i$ is the loss function for all the samples in device $i$ and $l$ is the loss function for a single sample. The goal of the FL

optimization problem can be formulated as follows:

$$\min_{w_{glb}} F(w_{glb}) = \frac{1}{|Dev|} \sum_{i=1}^{|Dev|} f_i(w_{glb}),$$

$$s.t., f_i(w_{glb}) = \frac{1}{n_i} \sum_{j=1}^{n_i} l(d_{i,j}; w_{glb}).$$

## 2.2. Related work on Heterogeneous FL

To address the low training performance caused by heterogeneous devices, existing methods attempt to dispatch heterogeneous models for local training. According to the structure of heterogeneous models, they can be classified into two categories, i.e., completely heterogeneous methods and sub-model extraction methods.

**Completely Heterogeneous Methods**. Completely heterogeneous methods typically dispatch models with completely different structures. Existing completely heterogeneous methods typically adopt knowledge distillation (KD) or prototypes. FedGKT (He et al., 2020) pioneers a grouped knowledge transfer framework where resource-constrained clients train lightweight models while leveraging aggregated knowledge from a larger server-side model. FedMD (Li & Wang, 2019) extends this by incorporating semi-supervised distillation using public proxy datasets to align predictions across heterogeneous clients, though reliance on public data raises privacy concerns. FedNTD (Lee et al., 2022) introduces noise-tolerant distillation to mitigate information loss during cross-model knowledge transfer, enhancing robustness in non-IID scenarios. FedProto (Tan et al., 2022) utilizes the intermediate outputs of models as prototypes and enables knowledge exchange by aligning prototypes of heterogeneous models. Although these methods enable model training among heterogeneous devices, due to without parameter sharing or relying on an additional dataset, the practices of these methods are seriously limited.

**Sub-model Extraction Methods**. Sub-model extraction methods generate heterogeneous models by pruning or partitioning a shared global model, ensuring parameter compatibility for aggregation. HeteroFL (Diao et al., 2020), AdaptiveFL (Jia et al., 2024), and FlexFL (Chen et al., 2024) introduce width-wise pruning to derive sub-models with varying channel dimensions, enabling federated training across devices with heterogeneous compute capabilities. ScaleFL (Ilhan et al., 2023) extends this by dynamically adjusting both width and depth proportions while incorporating skip connections to preserve feature compatibility between sub-models. DepthFL (Kim et al., 2023) adopts depth-wise pruning, removing deeper network layers to create shallower sub-models tailored for resource-constrained devices. AdaFL (Li et al., 2024) and AutoFL (Kim & Wu, 2021) leverage reinforcement learning (RL) to dynamically adjust sub-model sizes based on real-time device feedback, optimizing resource utilization. FedHM (Yao et al., 2021) and TDPFed (Wang et al., 2022) reduce communication overhead by decomposing model weights into low-rank tensors, though computational costs remain non-trivial for edge devices. FedRolex (Alam et al., 2022) cyclically activates different model blocks across training rounds, improving coverage and mitigating model drift. Based on FedRolex, FedBRB (Xu et al., 2024) and FedDSE (Wang et al., 2024a) explore dynamic block selection to prioritize critical modules, but their coarse-grained partitioning strategies neglect fine-grained channel-level optimizations, resulting in suboptimal resource utilization. However, since all these methods rely on fixed pruning ratios or static resource assumptions, limiting adaptability in uncertain scenarios. For instance, HeteroFL's uniform layer-wise pruning ignores activation sparsity patterns, while ScaleFL's rigid scaling factors fail to account for dynamic resource fluctuations.

To the best of our knowledge, SpineFL is the first attempt to adopt a layer-wise backbone sharing mechanism and a dynamic activation-guided neuron selection strategy to extract sub-models, which facilitates the generalization of the global model while alleviating the performance degradation caused by heterogeneous data among clients.

## 3. Our SpineFL Approach

### 3.1. Overview of Method

Figure 1 illustrates the framework and workflow of SpineFL, which consists of a central cloud server and multiple local devices. Different from the traditional FL, in each FL training round, the cloud server extracts a suitable sub-model for each selected device according to its resource and activation information from the global model and dispatches the extracted sub-models rather than the full global model for local training. As shown in Figure 1, the cloud server of SpineFL includes two key components, i.e., backbone generator and sub-model extractor. The backbone generator decomposes the global model into two parts, i.e., the backbone portion and the dynamic portion, according to the hardware resources of target devices. To enable the model generalization, SpineFL ensures that all the sub-models share the backbone portion. The sub-model extractor maintains two tables that store the resource information of local devices and their activation information. In each FL training round, the sub-model extractor selects part of the neurons in the dynamic selection portion according to the resource and activation of selected devices and all the neurons in the backbone portion to generate the sub-models. In practice, the available resource of a client can be estimated according to the model size that the client successfully uploads or reported proactively by the client device. The generated sub-models are dispatched to the corresponding local de-

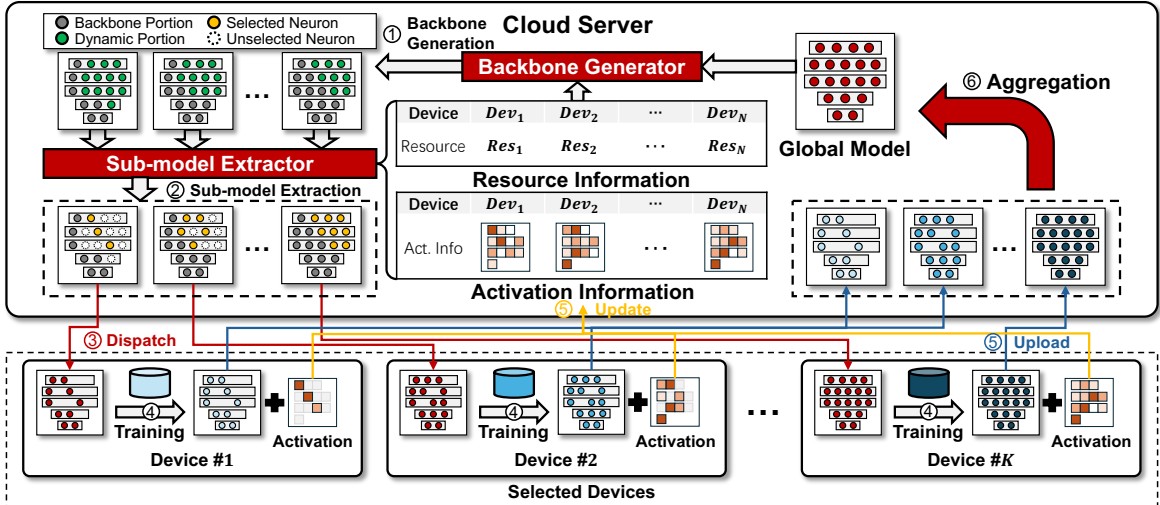

*Figure 1.* Framework and workflow of our SpineFL approach

vices for training. The workflow of our SpineFL in each FL training round is as follows:

- **Step 1: Backbone generation**: The cloud server randomly select $K$ devices as the activated clients for local training. For each selected device, the cloud server decomposes the global model into two portions, i.e., the backbone portion and the dynamic portion.

- **Step 2: Sub-Model Extraction**: For each selected device, the sub-model extractor selects neurons in the dynamic portion according to its resource and activation information and uses the selected neurons together with the corresponding backbone portion to generate the sub-model.

- **Step 3: Model Dispatching**: The cloud server dispatches the generated sub-models to local devices.

- **Step 4: Local Training**: Each activated client trains its received sub-model using its local data.

- **Step 5: Model and Activation Information Uploading**: Each client uploads its trained model and corresponding activation information to the cloud server.

- **Step 6: Model Aggregation**: The cloud server aggregates the corresponding parameters of the received local models to generate a new global model.

### 3.2. Implementation

Algorithm 1 presents the detailed workflow of our SpineFL approach. As shown in Algorithm 1, Lines 1-3 initialize the global model $w_0$, resource table $T_{res}$, and activation table $T_{act}$, respectively. Lines 4-16 present the FL training process, where Lines 5-15 show the details of each FL training

---

**Algorithm 1** The workflow of SpineFL

**Input:** i) $T$: Total communication rounds; ii) $S_d$: the set of devices; iii) $K$: number of activated devices in each FL training round.
**Output:** The trained global model $w_{glb}$
**SpineFL**($T$, $S_d$, $K$):

1: Initialize global model $w_0$
2: $T_{re} \leftarrow \texttt{ResInit}(S_d)$
3: $T_{act} \leftarrow \texttt{ActInit}(S_d)$
4: **for** $t = 0$ **to** $T - 1$ **do**
5:     $D_t \leftarrow$ Random select $K$ clients from $S_d$
6:     $S_m \leftarrow \{\}$
7:     **for** $d$ **in** $D_t$ **do**
8:         $w_{bc}, w_{dyn} \leftarrow \texttt{BackboneGen}(w_t, R_d)$
9:         $A_d \leftarrow T_{act}[d], R_d \leftarrow T_{res}[d]$
10:         $w_d \leftarrow \texttt{SubModExtract}(w_{dyn}, A_d, R_d, w_{bc})$
11:         $v_{t+1}^d, A_d, R_d \leftarrow \texttt{LocalTraining}(w_d)$
12:         $S_m \leftarrow S_m \cup \{v_{t+1}^i\}$
13:         $T_{act}[d] \leftarrow A_d, T_{res}[d] \leftarrow R_d$
14:     **end for**
15:     $w_{t+1} \leftarrow \texttt{ModelAggr}(S_m)$
16: **end for**
17: $w_{glb} \leftarrow w_T$
18: **return** $w_{glb}$

---

round. Line 5 randomly selects $K$ devices from $S_d$ as activated devices for local training. Line 6 initializes the local model pool $S_m$ used to collect trained local models. In Line 8, the function $\texttt{BackboneGen}(\cdot)$ decomposes the global model $w_t$ into the backbone portion $w_{bc}$ and the dynamic portion $w_{dyn}$ according to the hardware resources information $R_d$. In Line 10, the function $\texttt{SubModExtract}(\cdot)$ generates the sub-model for the device $d$. In Line 11, the sub-model $w_d$ is dispatched into the device $d$ for local training

---

**Algorithm 2** Layer-wise Backbone Generation

---

**Input:** i) $w$: the global model; ii)$R_d$: hardware resources of the target device $d$.
**Output:** i) $w_{bc}$: the backbone portion; ii) $w_{dyn}$: the dynamic portion.
**BackboneGen**($w$,$R_d$):

1: $L \leftarrow$ Number of layers for $w$
2: $[\alpha_1, \alpha_2, \ldots, \alpha_L] \leftarrow$ RatioCalculate($w, R_d$)
3: $w_{bc} \leftarrow$ Init($w$), $w_{dyn} \leftarrow$ Init($w$)
4: **for** $l = 1, \ldots, L$ **do**
5:     $n \leftarrow |w[l]|$
6:     $w_{bc}[l] \leftarrow w[l][1 : \lceil \alpha_l \times n \rceil]$
7:     $w_{dyn}[l] \leftarrow w[l][\lceil \alpha_l \times n \rceil + 1 : n]$
8: **end for**
9: **return** $w_{bc}, w_{dyn}$

---

and the device $d$ uploads the trained local model $v_{t+1}^d$, activation information $A_d$, and its current hardware resources information $R_d$. Line 13 updates the resource and activation table. In Line 15, the function ModelAggr($\cdot$) aggregates all the local models to generate a new global model $w_{t+1}$. After $T$ rounds of FL training, SpineFL returns the trained global model $w_{glb}$.

### 3.2.1. LAYER-WISE BACKBONE SHARING MECHANISM

To enhance global model generalization, SpineFL requires all sub-models to share a subset of parameters organized as a common backbone structure. In this way, the backbone can learn more generalized knowledge from all the devices. Specifically, SpineFL decomposes the global model into two portions, i.e., the backbone portion and the dynamic portion. During sub-model extraction, the extractor only prunes the dynamic portion and preserves the full backbone portion. Since the device heterogeneity, SpineFL assigns different size of backbones according to the hardware resources of the target devices rather than a fixed size for all the devices. Specifically, to ensure that the resource-constrained devices can fully train all the parameters, SpineFL assigns a small size of backbone. To ensure model generalization, for devices with high hardware resources, SpineFL assigns a large size of backbone. Note that each backbone is a subset of a larger backbone. In addition, according to the observations in (Jia et al., 2024; Li et al., 2017) that pruning the neurons in deep layers can cause less performance degradation than pruning those of shallow layers, SpineFL prefers to assign more parameters in shallow layers as backbones, which will not be pruned in the sub-model extraction process.

Algorithm 2 presents the layer-wise backbone generation process. In Line 2, the function RatioCalculate($\cdot$) calculates the ratio of backbone for each layer in the global model $w$ according to the hardware resources $R_d$, where

$\alpha_i \in [0, 1.0]$ is the ratio for the backbone in the $i$th layer. Note that for devices with different hardware resources, the RatioCalculate($\cdot$) will generate different settings of backbone ratios. Note that for devices $d_i$ and $d_j$, if $R_{d_i} > R_{d_j}$, for each layer $l$, $\alpha_l^{d_i} \geq \alpha_l^{d_j}$. In addition, for the $i$th and $j$th layer, if $i > j$, then $\alpha_i \leq \alpha_j$. Line 3 initializes the backbone portion $w_{bc}$ and the dynamic portion $w_{dyn}$ based on $w$. Lines 4-8 shows the generation process for the backbone portion and the dynamic portion. Specifically, for the $l$th layer, Line 6 assigns the first $\lceil \alpha_l \times n \rceil$ neurons to the $l$th layer of $w_{bc}$ and Line 7 assigns the remaining neurons to the $l$th layer of $w_{dyn}$.

### 3.2.2. ACTIVATION-GUIDED NEURON SELECTION STRATEGY

To alleviate the gradient conflict problem caused by data heterogeneity, SpineFL adopts the activation as a metric to select the neurons in the dynamic portion for sub-model extraction. Specifically, SpineFL records and updates the activation information of all devices on the server side at the end of each local training stage. Crucially, during the sub-model extraction for a target device in the current round $t$, SpineFL utilizes the historical activation values compiled from the previous round $t - 1$ rather than computing new ones in the current round. This design ensures that sub-model extraction imposes no additional computational overhead on either the server or the client before local deployment. Based on these historical footprints, SpineFL prefers to select neurons with higher cumulative activation values.

Algorithm 3 presents the workflow of the sub-model extraction process. Line 2 computes the number of neurons to be selected according to the hardware resource of the target device, where $n_i$ denotes the number of neurons selected in the $i$th layer. Line 3 initializes the sub-model $w_d$. In Line 5, the function ScoreCal($\cdot$) calculates the importance score of each neuron in the $l$th layer of the dynamic portion according to their activation information. In Line 6, the function NeuronSelect($\cdot$) selects the neurons based on the calculated scores $S_l$. Line 7 generates the $l$th layer of the sub-model by combing the neurons in the $l$th layer of backbone portion and selected neurons $lw_l$.

**Score Calculation**. For a target device $d$, we perform a forward pass using the local data $\mathcal{D}_k^t$ through the local model $w_d$ to obtain feature maps $F_A$. For each neuron $i$ in layer $l$, we compute its importance score $s_i$ based on the average absolute activation values across all spatial positions:

$$s_{l,i} = \frac{1}{H \cdot W} \sum_{j=1}^{H} \sum_{k=1}^{W} |A_d[l][i, j, k]|,$$

where $A_d[l][i, j, k]$ denotes the activation value at position $(j, k)$ of neuron $i$ in the $l$th layer, and $H \times W$ is the spatial size of the feature map.

**Algorithm 3** Sub-model Extraction

**Input:** i) $w_{dyn}$: the dynamic portion; ii) $A_d$: the activation information for device $d$; iii) $R_d$: the hardware resource information of device $d$; iv) $w_{bc}$: the backbone portion.
**Output:** $w_d$: the extracted sub-model for device $d$
**SubModExtract**($w_{dyn}$, $A_d$, $R_d$, $w_{bc}$):

1: $L \leftarrow$ Number of layers for $w_{dyn}$
2: $[n_1, n_2, \ldots, n_L] \leftarrow$ NeuronNumCal($w_{dyn}$, $R_d$)
3: $w_d \leftarrow$ Init($w_{bc}$)
4: **for** $l = 1, \ldots, L$ **do**
5:    $S_l \leftarrow$ ScoreCal($w_{dyn}[l]$, $A_d[l]$)
6:    $lw_l \leftarrow$ NeuronSelect($w_{dyn}$, $P_l$, $n_l$)
7:    $w_d[l] \leftarrow [w_{bc}[l]; lw_l]$
8: **end for**
9: **return** $w_d$

**Neuron Selection.** Based on the calculated score, SpineFL selects the neurons in the dynamic portion. To alleviate the gradient conflict, SpineFL prefers to select the neurons with a high importance score. To ensure that each device can fully train all the parameters, SpineFL selects neurons with the probability as follows:

$$P_{l,i} = \frac{\exp(s_{l,i}/T)}{\sum_{j=1}^{|l|} \exp(s_{l,j}/T)},$$

where $T$ is the temperature to control the impact of the score for neuron selection. When $T \to \infty$, the selection becomes nearly uniform across all neurons; conversely, when $T \to 0$, SpineFL prefers to select neurons with the highest score. In this way, SpineFL can select neurons with a high importance score with a higher probability.

### 3.2.3. MODEL AGGREGATION.

Similar to the traditional FL methods, SpineFL aggregates all the local models to generate the global models. Assume that $S_m$ is the list of local models and $n_{l,i}^j$ is the $i$th neuron in the layer $l$ in the $j$th model in $S_m$. The corresponding neuron of the global model can be generated as follows:

$$n_{l,i}^{glb} = \frac{\sum_{j=1}^{|S_m|} \text{GetNeuron}(S_m[j], l, i)}{\sum_{j=1}^{|S_m|} \text{IsSel}(S_m[j], l, i)},$$

where the function GetNeuron($S_m[j]$, $l$, $i$) returns the neuron $n_{l,i}^j$ if the model $S_m[j]$ includes the corresponding neuron. Otherwise, the function returns the initial neuron where all the elements are zero. The function IsSel($S_m[j]$, $l$, $i$) returns 1, if the model $S_m[j]$ includes the neuron $n_{l,i}^j$. Otherwise, the function returns 0.

## 4. Convergence Analysis

We analyze the convergence of SpineFL based on three foundational assumptions aligned with federated optimization (Li et al., 2020) and neural network dynamics:

**Assumption 4.1 (L-Smooth Objective):** Each local loss function $F_n(\mathbf{w})$ is $L$-smooth:

$$\forall \mathbf{w}, \mathbf{w}' \in \mathbb{R}^d, \quad \|\nabla F_n(\mathbf{w}) - \nabla F_n(\mathbf{w}')\| \leq L\|\mathbf{w} - \mathbf{w}'\|.$$

**Assumption 4.2 (Bounded Gradient Variance):** The stochastic gradient $g_n(\mathbf{w})$ satisfies:

$$\mathbb{E}[\|g_n(\mathbf{w})\|_2^2] \leq G^2, \quad \mathbb{E}[\|g_n(\mathbf{w}) - \nabla F_n(\mathbf{w})\|_2^2] \leq \sigma^2.$$

**Assumption 4.3 (Bounded Gradient Norm):** For all $\mathbf{w}$ and $n \in [N]$, there exists $G_f > 0$ such that:

$$\|\nabla F_n(\mathbf{w})\|_2 \leq G_f.$$

**Lemma 4.4 (Gradient Error Bound):** Let $\nabla f_n(\mathbf{w}; x)$ and $\nabla \hat{f}_n(\mathbf{w}; x)$ be the full-model and sub-model gradients respectively. Then:

$$\|\nabla f_n(\mathbf{w}; x) - \nabla \hat{f}_n(\mathbf{w}; x)\|_2 \leq G_h^2 H^2 \sum_{l,i} \|e_{l,i,k}\|_2,$$

where $e_{l,i,k}$ is the activation error of the $i$-th neuron in layer $l$, $H$ is the network depth, and $G_h$ bounds hidden layer activations.

**Lemma 4.5 (Activation Error Bound):** For any layer $l$ and retention ratio $r$, there exists $C_l > 0$ such that:

$$\mathbb{E}_x \left[\|h_l^{(r)}(x) - h_l(x)\|_2^2\right] \leq C_l(1 - r).$$

**Theorem 1 (SpineFL Convergence):** Under Assumptions 4.1–4.3 and Lemmas 4.4–4.5, after $T$ iterations with learning rate $\eta_t = \frac{\mu}{t+\nu}$ for $\mu, \nu > 0$:

$$\frac{1}{T} \sum_{t=1}^{T} \mathbb{E}\left[\|\nabla F(\mathbf{w}^t)\|_2^2\right] \leq \frac{C_1}{\eta T} + C_2(1 - r) + \eta C_3,$$

where $C_1 = \frac{\sigma^2}{\mu} + \frac{LD^2}{\mu}$, $C_2 = G_h^4 H^4 \sum_l C_l$, $C_3 = \frac{LG^2}{2}$, $\mu$ is the compression factor, and $D$ bounds parameter updates.

The full proofs are presented in Appenix A.

## 5. Experiments

To evaluate the performance of SpineFL, we conducted extensive experiments on well-known datasets and underlying

*Table 1.* Test accuracy under different data distributions (IID, Non-IID $\alpha = 0.6$, $\alpha = 0.3$)

| Model | Dataset | Distribution | Method | | | | | | | | |
|---|---|---|---|---|---|---|---|---|---|---|---|
| | | | HeteroFL | FedRolex | FedDSE | ScaleFL | FlexFL | Fjord | AdaptiveFL | FIARSE | SpineFL(Ours) |
| ResNet-34 | EMNIST | IID | 94.32±0.51 | 96.75±0.43 | 98.50±0.38 | 98.21±0.42 | 98.92±0.20 | 95.37±0.36 | 97.69±0.21 | 98.31±0.23 | **99.51±0.34** |
| | | α=0.6 | 92.93±0.77 | 94.69±0.87 | 97.36±1.02 | 96.34±0.97 | 97.01±0.66 | 92.83±0.89 | 95.19±0.74 | 96.65±0.84 | **98.15±0.52** |
| | | α=0.3 | 90.21±1.14 | 93.41±0.91 | 96.06±1.29 | 93.96±1.45 | 95.15±1.32 | 91.07±1.16 | 94.29±1.04 | 95.31±0.98 | **98.14±0.93** |
| | CIFAR-10 | IID | 78.32±0.52 | 79.28±0.91 | 83.59±0.74 | 80.33±0.34 | 84.70±0.42 | 79.54±0.32 | 84.12±0.41 | 84.85±0.28 | **86.17±0.19** |
| | | α=0.6 | 71.26±0.79 | 72.31±1.54 | 80.77±1.22 | 78.40±0.89 | 79.75±0.88 | 72.07±0.63 | 78.32±1.16 | 78.76±1.25 | **82.46±0.54** |
| | | α=0.3 | 63.03±0.95 | 64.26±1.87 | 76.64±1.76 | 71.07±1.44 | 76.40±1.22 | 64.25±1.08 | 74.59±1.37 | 73.88±1.42 | **78.30±1.46** |
| | CIFAR-100 | IID | 38.07±0.53 | 40.64±1.21 | 48.04±0.14 | 45.52±0.13 | 47.02±0.67 | 39.42±0.31 | 47.64±0.38 | 47.21±0.52 | **50.31±0.72** |
| | | α=0.6 | 32.01±1.34 | 36.83±1.49 | 46.71±1.25 | 43.90±1.31 | 45.90±1.07 | 34.78±0.52 | 44.39±1.06 | 43.51±1.25 | **48.13±0.87** |
| | | α=0.3 | 27.94±2.45 | 28.86±2.01 | 40.63±1.87 | 33.79±1.95 | 33.56±1.90 | 29.14±1.07 | 40.13±1.39 | 29.14±1.07 | **43.46±1.28** |
| | TinyImageNet | IID | 22.54±0.36 | 20.84±0.22 | 34.79±0.37 | 24.39±0.49 | 29.64±0.21 | 23.24±0.41 | 32.78±0.39 | 31.47±0.32 | **36.51±0.36** |
| | | α=0.6 | 18.87±0.66 | 17.47±0.83 | 30.27±0.41 | 20.45±1.23 | 25.70±0.82 | 19.26±0.62 | 29.84±0.85 | 28.74±0.69 | **31.57±0.61** |
| | | α=0.3 | 15.42±1.10 | 15.73±1.14 | 25.50±1.17 | 19.91±0.87 | 23.71±1.48 | 16.43±1.01 | 24.39±1.24 | 24.17±0.93 | **28.59±1.32** |
| CNN | EMNIST | IID | 94.74±0.26 | 97.36±0.27 | 98.63±0.26 | 96.86±0.25 | 98.24±0.31 | 94.36±0.27 | 97.51±0.32 | 97.58±0.22 | **98.72±0.19** |
| | | α=0.6 | 93.29±0.49 | 95.29±0.53 | 96.85±0.69 | 95.04±0.46 | 96.42±0.47 | 93.14±0.51 | 95.04±0.67 | 95.31±0.62 | **97.67±0.42** |
| | | α=0.3 | 90.54±0.92 | 94.01±0.88 | 94.32±1.08 | 92.76±0.97 | 94.72±0.85 | 91.26±0.89 | 93.84±0.92 | 93.17±1.03 | **96.01±0.74** |
| | CIFAR-10 | IID | 66.28±0.57 | 69.09±0.46 | 81.37±0.97 | 79.48±0.75 | 81.71±0.86 | 67.34±0.42 | 82.95±0.57 | 83.12±0.71 | **84.67±0.53** |
| | | α=0.6 | 65.13±1.86 | 66.68±1.84 | 80.19±1.58 | 76.39±1.34 | 79.14±1.77 | 64.76±0.89 | 80.48±1.07 | 81.06±1.21 | **82.29±1.35** |
| | | α=0.3 | 60.47±2.49 | 63.34±2.04 | 78.83±2.20 | 74.99±2.07 | 76.78±2.16 | 61.08±1.76 | 76.42±1.93 | 78.47±2.04 | **80.31±1.67** |
| | CIFAR-100 | IID | 33.13±1.69 | 41.83±0.47 | 49.43±0.49 | 44.51±0.45 | 48.50±0.51 | 33.67±1.13 | 47.93±0.63 | 50.14±0.57 | **52.56±0.59** |
| | | α=0.6 | 32.14±1.76 | 38.86±1.32 | 48.54±1.04 | 43.04±0.91 | 47.35±0.88 | 31.50±1.47 | 45.31±1.12 | 47.76±0.87 | **51.47±0.83** |
| | | α=0.3 | 31.45±2.43 | 37.71±2.12 | 46.85±1.54 | 42.76±1.72 | 46.02±1.46 | 28.97±2.01 | 43.73±1.67 | 46.29±1.52 | **50.13±1.61** |
| | TinyImageNet | IID | 22.09±1.21 | 25.06±0.80 | 28.35±0.54 | 25.86±0.43 | 27.03±0.52 | 22.47±0.97 | 26.73±0.45 | 28.51±0.49 | **30.15±0.38** |
| | | α=0.6 | 21.12±1.69 | 23.46±1.25 | 26.23±0.84 | 24.62±0.93 | 26.36±0.64 | 20.61±1.26 | 24.13±0.71 | 26.58±0.86 | **29.04±0.50** |
| | | α=0.3 | 19.86±2.02 | 21.54±1.57 | 25.11±1.07 | 25.24±1.21 | 25.91±0.93 | 18.93±1.51 | 21.76±1.12 | 24.71±1.14 | **27.43±0.78** |
| VGG-16 | EMNIST | IID | 94.95±0.14 | 97.67±0.19 | 98.53±0.12 | 96.19±0.16 | 97.91±0.15 | 95.11±0.20 | 96.85±0.18 | 97.34±0.16 | **98.33±0.11** |
| | | α=0.6 | 93.47±0.35 | 95.59±0.36 | 97.46±0.25 | 94.34±0.62 | 96.26±0.37 | 93.81±0.37 | 95.14±0.38 | 95.93±0.30 | **97.59±0.23** |
| | | α=0.3 | 90.71±0.84 | 94.30±0.86 | 95.74±0.44 | 92.16±0.73 | 94.50±0.59 | 91.06±0.74 | 93.75±0.53 | 93.43±0.61 | **96.45±0.54** |
| | CIFAR-10 | IID | 71.58±0.25 | 68.97±0.52 | 83.16±0.46 | 75.22±0.28 | 81.68±0.57 | 73.41±0.42 | 84.62±0.36 | 82.77±0.41 | **88.32±0.34** |
| | | α=0.6 | 65.34±0.74 | 66.80±1.23 | 80.22±0.59 | 75.35±0.43 | 79.87±0.81 | 68.04±0.79 | 80.19±0.62 | 79.71±0.80 | **85.13±0.56** |
| | | α=0.3 | 64.94±1.62 | 63.16±1.99 | 78.69±1.67 | 74.36±2.13 | 77.74±1.43 | 64.55±1.33 | 77.46±1.12 | 77.05±1.57 | **84.34±1.83** |
| | CIFAR-100 | IID | 34.42±1.13 | 41.04±1.32 | 49.55±0.48 | 43.59±0.63 | 48.87±0.39 | 33.58±1.03 | 48.63±0.72 | 48.81±0.69 | **52.30±0.33** |
| | | α=0.6 | 32.94±1.59 | 40.08±1.63 | 48.71±0.68 | 42.14±0.83 | 47.62±0.55 | 31.18±1.34 | 46.51±1.04 | 46.71±0.94 | **51.19±0.52** |
| | | α=0.3 | 30.65±1.76 | 35.03±2.14 | 47.18±1.01 | 38.27±1.15 | 42.82±1.57 | 29.72±1.57 | 43.71±1.28 | 44.17±1.53 | **49.76±1.47** |
| | TinyImageNet | IID | 23.47±1.57 | 25.79±0.74 | 29.32±0.89 | 26.02±0.43 | 27.51±0.63 | 24.67±1.02 | 27.34±0.56 | 28.93±0.47 | **32.51±0.72** |
| | | α=0.6 | 21.81±1.92 | 22.40±1.36 | 27.82±1.37 | 24.73±0.94 | 26.18±0.85 | 22.18±1.32 | 24.78±0.96 | 26.54±0.99 | **30.13±0.94** |
| | | α=0.3 | 18.85±2.24 | 19.44±1.56 | 25.68±1.49 | 23.68±1.32 | 24.71±1.19 | 19.75±1.67 | 22.41±1.45 | 23.79±1.54 | **27.94±1.67** |

DNN models. The subsequent subsections aim to answer the following three research questions (RQs).

**RQ1: (Superiority of SpineFL)**: How does SpineFL outperform state-of-the-art methods?

**RQ2: (Compatibility of SpineFL)**: What is the performance of SpineFL with different settings (e.g., client data distributions, DNN architectures, datasets)?

**RQ3: (Benefits of SpineFL Components)**: Can our proposed techniques improve classification performance?

## 5.1. Experimental Settings

To evaluate the performance of SpineFL, we implemented SpineFL using PyTorch. For all the investigated FL methods, we adopted the same SGD optimizer with a learning rate of 0.001 and a momentum of 0.9. For local training, we set the batch size to 32 and the local epoch to 3. We assume a total of $|Dev| = 100$ AIoT devices participate in the training, with 10% of them ($f = 10\%$) randomly selected in each FL round. All experiments were conducted on an Ubuntu workstation equipped with an Intel i9-13900K CPU, 32 GB of RAM, and an NVIDIA RTX 4090 GPU.

**Dataset Settings**: We conducted experiments on four well-

known datasets, i.e., CIFAR-10, CIFAR-100, EMNIST, and TinyImageNet. To evaluate the performance of SpineFL within both IID and non-IID scenarios, we adopted the Dirichlet distribution (Hsu et al., 2019) denoted by $\alpha$ to control the heterogeneity settings.

**Model Settings**: We investigated three well-known models, i.e., CNN, ResNet-34, VGG-16. The structure of CNN model was same as the model used in (McMahan et al., 2017). ResNet-34 and VGG-16 models were obtained from the official library (Contributors, 2024).

**Baseline Methods**: To assess the effectiveness of SpineFL, we compare it against five well-established baselines, comprising the classical HeteroFL (Diao et al., 2020) and four state-of-the-art FL optimization methods, i.e., FedRolex (Alam et al., 2022), FedDSE (Wang et al., 2024a), ScaleFL (Ilhan et al., 2023), FlexFL (Chen et al., 2024), Fjord (Horvath et al., 2021), and AdaptiveFL (Jia et al., 2024), FIARSE (Wu et al., 2024). Please see C.1 for the details of the baselines.

**Device Settings**: We present three types of devices, i.e., high-quality device, medium-quality device, and low-quality device, respectively, where the high-quality device can train the fully global model, the medium-quality device can only train the model with a 60% size of the global model, and

the low-quality device can only train the model with a 30% size of the global model.

## 5.2. Performance Comparison(RQ1)

Table 1 shows the test accuracy of all methods across four datasets under varying data distributions (IID and Non-IID settings), where the first column indicates the type of global model, the second column denotes the type of dataset, the third column shows the configuration of data heterogeneity, and the forth column has nine sub-columns, which shows the test accuracy of eight baselines and our SpineFL approach.

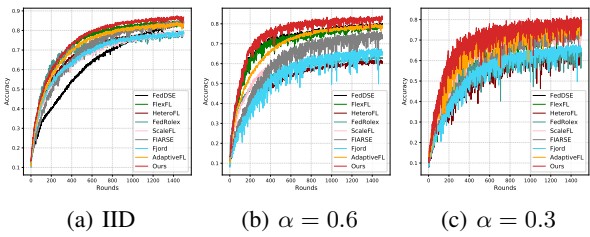

*Figure 2.* Learning curves of different FL methods using ResNet-34 on CIFAR-10

From Table 1, we can observe that our SpineFL can achieve the highest test accuracy on all the cases. As shown in Table 1, in the most challenging Non-IID setting ($\alpha = 0.3$), our approach significantly outperforms the strongest baseline, FedDSE, by clear margins across all datasets. Figure 2 presents learning curves of all the nine methods with CIFAR-10 dataset using ResNet-34 model. We can observe that SpineFL can outperform all the baselines when they achieve their highest test accuracy.

## 5.3. Compatibility Analysis (RQ2)

### 5.3.1. CONFIGURATIONS OF HARDWARE RESOURCES

To demonstrate the effectiveness of SpineFL with different configurations of hardware resources, we conducted experiments on three configurations, i.e., low resource, medium-resource, and high-resource, respectively, where the low-resource configuration consists of 50% low-quality devices, 30% medium-quality devices, and 20% high-quality devices. The medium-resource configuration consists of 33% low-quality devices, 34% medium-quality devices, and 33% high-quality devices. The high-resource configuration consists of 20% low-quality devices, 30% medium-quality devices, and 50% high-quality devices. Table 2 presents the experimental results of nine FL methods with three resource configurations. We can find that SpineFL can achieve the best accuracy in all the resource configurations.

*Table 2.* Test accuracy (%) under different resource situation

| Method | low-resource | medium-resource | high-resource |
|---|---|---|---|
| HeteroFL | 57.54±1.68 | 71.26±0.79 | 74.91±1.57 |
| FedRolex | 58.54±1.61 | 72.31±1.87 | 74.35±2.37 |
| FedDSE | 76.78±1.23 | 80.77±1.22 | 82.69±1.13 |
| ScaleFL | 75.42±1.39 | 78.40±0.89 | 82.22±1.43 |
| FlexFL | 76.25±1.26 | 79.75±0.88 | 83.71±1.58 |
| Fjord | 56.36±0.98 | 72.07±0.63 | 74.31±1,32 |
| AdaptiveFL | 73.04±1.34 | 78.32±1.16 | 81.12±1.49 |
| FIARSE | 74.61±1.05 | 78.76±1.25 | 82.03±1.51 |
| **SpineFL** | **77.83±1.13** | **82.46±0.54** | **84.32±0.56** |

### 5.3.2. NUMBER OF INVOLVED DEVICES

To evaluate the effectiveness of SpineFL with different numbers of involved devices, we conduct experiments for all the nine methods with the number of total devices $|S_d| = 100, 200, 500$, respectively. Figure 4 shows that SpineFL can achieve the highest accuracy compared with that of all the baselines. In addition, we can observe that fewer total devices can use fewer training rounds for convergence. This is mainly because the high number of total devices results in few samples in each device.

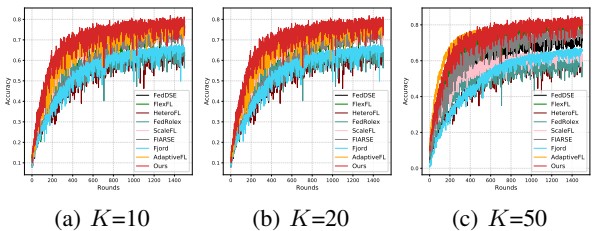

*Figure 3.* Learning curves of nine methods for different numbers of activated clients on CIFAR-10 dataset with $\alpha = 0.3$

### 5.3.3. NUMBER OF SELECTED DEVICES

To evaluate the effectiveness of SpineFL with different numbers of selected devices, we conduct experiments for all the nine methods with the number of selected devices $K = 10, 20, 50$, respectively. Figure 3 presents the corresponding results. From Figure 3, we can find that SpineFL still outperforms all the baselines in all the cases. We can also find that fewer number of selected devices leads to slower initial convergence but eventually reaches similar final accuracy. For example, $K = 50$ achieves 0.8 accuracy within 400 rounds, while $k = 10$ converges to about 0.83 after 1000 rounds.

## 5.4. Ablation study (RQ3)

To test the performance of the neuron selection and backbone strategy, we conducted experiments on the CIFAR-10 dataset under IID and non-IID ($\alpha$=0.3, 0.6) scenarios. We

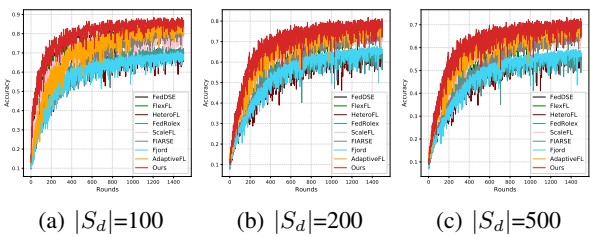

(a) $|S_d|$=100  (b) $|S_d|$=200  (c) $|S_d|$=500

*Figure 4.* Learning curves of nine methods for different numbers of clients on CIFAR-10 dataset with $\alpha = 0.3$

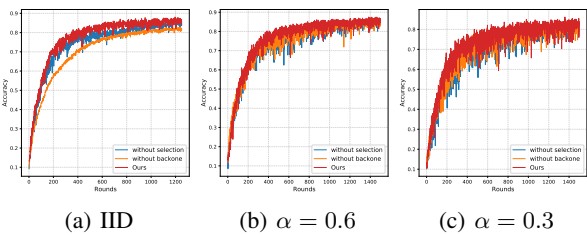

(a) IID  (b) $\alpha = 0.6$  (c) $\alpha = 0.3$

*Figure 5.* Ablation study

utilized three model selection strategies, where "without selection" randomly chooses activation neurons while keeping the backbone. We utilized three model selection strategies, where "without backbone" deletes the backbone mechanism. As shown in Figure 5, the origin method consistently achieves the best performance. For instance, in the IID scenario, it attains an accuracy of 86.67%, outperforming the random selection strategy (85.25%) and the variant without backbone (83.15%).

## 6. Conclusion

This paper proposes a resource-adaptive FL framework that addresses the limitations of existing sub-model extraction methods in heterogeneous environments. By integrating the backbone with dynamic identification of representative submodules, our approach enables precise and efficient model adaptation to client-specific resources and data characteristics. It ensures high-utility neuron selection through activation analysis and supports scalable model customization, facilitating fair and effective participation across diverse devices. Extensive experiments show that the method significantly enhances global model performance, especially for low-resource clients and complex data distributions. It also accelerates convergence and reduces communication costs, offering a practical and scalable solution for FL systems.

## Acknowledgements

This research is supported by the National Natural Science Foundation of China (Grant Nos. 62572244, 62002047), the Sichuan Regional Innovation Cooperation Project (Grant No. 2025YFHZ0302), the National Key Laboratory of Aircraft Fluid Physics Fund (Grant No. 2025-APF-KFMS-21), and the National Research Foundation, Singapore, and Cyber Security Agency of Singapore under its National Cybersecurity R&D Programme and CyberSG R&D Cyber Research Programme Office. Any opinions, findings and conclusions or recommendations expressed in this material are those of the author(s) and do not reflect the views of National Research Foundation, Singapore, Cyber Security Agency of Singapore as well as CyberSG R&D Programme Office, Singapore. Ming Hu is the corresponding author.

## Impact Statement

This paper presents a heterogeneous federated learning approach named SpineFL to improve the performance of federated learning in resource-constrained scenarios. Our approach effectively promotes the application of federated learning in AIoT systems and cloud systems. Since SpineFL is still based on the traditional federated learning architecture, it does not result in any new ethical concerns.

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

# A. Proof

## A.1. Proof of convergence analysis

### A.1.1. PROOF OF LEMMA 4.4

The gradient error introduced by submodel training is proportional to the activation error, i.e.,

$$\|\nabla f_n(w;x) - \nabla \hat{f}_n(w;x)\|_2 \le G_h^2 H^2 \sum_{l,i} \|e_{l,i,k}\|_2,$$

where $\nabla f_n(w;x)$ is the full model gradient, $\nabla \hat{f}_n(w;x)$ is the submodel gradient, $e_{l,i,k}$ is the activation error of the $i$-th neuron in the $l$-th layer, $G_h$ is the upper bound of hidden layer activations and $H$ is the network depth.

Let the output of the full model be $f_n(w;x)$, and the output of the submodel be $\hat{f}_n(w;x)$. According to the chain rule of backpropagation, the gradient of the full model can be decomposed as:

$$\nabla f_n(w;x) = \sum_{l=1}^{H} \frac{\partial f_n}{\partial h_l} \cdot \frac{\partial h_l}{\partial w},$$

where $h_l$ is the activation value of the $l$-th layer, $\frac{\partial f_n}{\partial h_l}$ is the partial derivative of the output with respect to the activation value of the $l$-th layer, and $\frac{\partial h_l}{\partial w}$ is the partial derivative of the activation value of the $l$-th layer.

Similarly, the gradient of the submodel is:

$$\nabla \hat{f}_n(w;x) = \sum_{l=1}^{H} \frac{\partial f_n}{\partial \hat{h}_l} \cdot \frac{\partial \hat{h}_l}{\partial w},$$

where $\hat{h}_l = h_l - e_l$ is the activation value after pruning, and $e_l$ is the activation error of the $l$-th layer.

The gradient error is given by:

$$\|\nabla f_n - \nabla \hat{f}_n\|_2 = \left\| \sum_{l=1}^{H} \left( \frac{\partial f_n}{\partial h_l} \cdot \frac{\partial h_l}{\partial w} - \frac{\partial f_n}{\partial \hat{h}_l} \cdot \frac{\partial \hat{h}_l}{\partial w} \right) \right\|_2.$$

According to the triangle inequality, it can be decomposed into the sum of contributions from each layer:

$$\|\nabla f_n - \nabla \hat{f}_n\|_2 \le \sum_{l=1}^{H} \left\| \frac{\partial f_n}{\partial h_l} \cdot \frac{\partial h_l}{\partial w} - \frac{\partial f_n}{\partial \hat{h}_l} \cdot \frac{\partial \hat{h}_l}{\partial w} \right\|_2.$$

For each layer $l$, define:

$$\Delta_l = \left\| \frac{\partial f_n}{\partial h_l} \cdot \frac{\partial h_l}{\partial w} - \frac{\partial f_n}{\partial \hat{h}_l} \cdot \frac{\partial \hat{h}_l}{\partial w} \right\|_2.$$

Decompose it into two parts:

$$\Delta_l \le \left\| \left( \frac{\partial f_n}{\partial h_l} - \frac{\partial f_n}{\partial \hat{h}_l} \right) \cdot \frac{\partial h_l}{\partial w} \right\|_2 + \left\| \frac{\partial f_n}{\partial \hat{h}_l} \cdot \left( \frac{\partial h_l}{\partial w} - \frac{\partial \hat{h}_l}{\partial w} \right) \right\|_2.$$

Consider $\frac{\partial f_n}{\partial h_l} - \frac{\partial f_n}{\partial \hat{h}_l}$. Since $\hat{h}_l = h_l - e_l$, according to Taylor's expansion, the difference in the derivative of the output with respect to $h_l$ can be expressed as:

$$\frac{\partial f_n}{\partial h_l} - \frac{\partial f_n}{\partial \hat{h}_l} = \frac{\partial^2 f_n}{\partial h_l^2} \cdot e_l + o(\|e_l\|).$$

Assume that the function $f_n$ is $L$-Lipschitz continuous with respect to $h_l$, then:

$$\left\| \frac{\partial f_n}{\partial h_l} - \frac{\partial f_n}{\partial \hat{h}_l} \right\|_2 \le L \cdot \|e_l\|_2.$$

Furthermore, assuming the upper bound of hidden layer activations is $G_h$, then:

$$\left\| \frac{\partial f_n}{\partial h_l} \cdot \frac{\partial h_l}{\partial w} - \frac{\partial f_n}{\partial \hat{h}_l} \cdot \frac{\partial \hat{h}_l}{\partial w} \right\|_2 \le L \cdot \|e_l\|_2 \cdot \left\| \frac{\partial h_l}{\partial w} \right\|_2.$$

Consider $\frac{\partial h_l}{\partial w} - \frac{\partial \hat{h}_l}{\partial w}$. Since $\hat{h}_l = h_l - e_l$, its derivative with respect to $w$ is:

$$\frac{\partial \hat{h}_l}{\partial w} = \frac{\partial h_l}{\partial w} - \frac{\partial e_l}{\partial w}.$$

Therefore, the difference term is:

$$\left\| \frac{\partial h_l}{\partial w} - \frac{\partial \hat{h}_l}{\partial w} \right\|_2 = \left\| \frac{\partial e_l}{\partial w} \right\|_2.$$

Assume that the upper bound of $\left\| \frac{\partial e_l}{\partial w} \right\|$ is $G_h \cdot \|e_l\|_2$, then:

$$\left\| \frac{\partial f_n}{\partial \hat{h}_l} \cdot \left( \frac{\partial h_l}{\partial w} - \frac{\partial \hat{h}_l}{\partial w} \right) \right\|_2 \le \left\| \frac{\partial f_n}{\partial \hat{h}_l} \right\|_2 \cdot G_h \cdot \|e_l\|_2.$$

Since $\left\| \frac{\partial f_n}{\partial \hat{h}_l} \right\|_2 \le G_h$, therefore:

$$\left\| \frac{\partial f_n}{\partial \hat{h}_l} \cdot \left( \frac{\partial h_l}{\partial w} - \frac{\partial \hat{h}_l}{\partial w} \right) \right\|_2 \le G_h^2 \cdot \|e_l\|_2.$$

Combine the two parts to obtain the upper bound of the single-layer gradient error:

$$\Delta_l \le (L + G_h^2) \cdot \|e_l\|_2.$$

Since the network depth is $H$, the total gradient error is:

$$\|\nabla f_n - \nabla \hat{f}_n\|_2 \le \sum_{l=1}^{H} (L + G_h^2) \cdot \|e_l\|_2.$$

Assuming $L \leq G_h^2$, then:

$$\|\nabla f_n - \nabla \hat{f}_n\|_2 \leq 2G_h^2 H \sum_{l=1}^{H} \|e_l\|_2.$$

Decompose the layer-wise activation error $\|e_l\|_2$ further into the sum of neuron-level errors $\|e_{l,i,k}\|_2$:

$$\sum_{l=1}^{H} \|e_l\|_2 \leq \sum_{l=1}^{H} \sum_{i,k} \|e_{l,i,k}\|_2.$$

Therefore, the upper bound of the total gradient error is:

$$\|\nabla f_n - \nabla \hat{f}_n\|_2 \leq 2G_h^2 H^2 \sum_{l,i,k} \|e_{l,i,k}\|_2.$$

In summary, the gradient error introduced by submodel training satisfies:

$$\|\nabla f_n(w; x) - \nabla \hat{f}_n(w; x)\|_2 \leq G_h^2 H^2 \sum_{l,i} \|e_{l,i,k}\|_2,$$

where $G_h$ is the upper bound of hidden layer activations, $H$ is the network depth, and $e_{l,i,k}$ is the activation error of the $i$-th neuron in the $l$-th layer.

A.1.2. PROOF OF LEMMA 4.5

For any layer $l$, the squared norm of the activation error satisfies:

$$\mathbb{E}_x \left[ \|h_l^{(r)}(x) - h_l(x)\|_2^2 \right] \leq C_l(1 - r),$$

where:

- $h_l(x)$ is the activation value of the $l$-th layer in the full model;

- $h_l^{(r)}(x)$ is the activation value of the submodel after pruning, retaining $r \cdot d_l$ largest activation values;

- $C_l$ is a constant related to the network structure;

- $r \in (0, 1)$ is the retention ratio.

The pruning operation retains the $r \cdot d_l$ activation values with the largest absolute values in layer $l$, setting the rest to 0. The error is defined as:

$$\|h_l^{(r)}(x) - h_l(x)\|_2^2 = \sum_{i \in \text{pruned indices}} h_{l,i}(x)^2.$$

Assume $h_{l,1}(x) \geq h_{l,2}(x) \geq \cdots \geq h_{l,d_l}(x) \geq 0$ (assuming non-negative activation values, such as ReLU). The pruning operation retains the first $k = \lfloor r \cdot d_l \rfloor$ largest values, setting

the remaining $d_l - k$ values to 0. Therefore, the pruned activation values satisfy:

$$h_{l,k+1}(x) \leq h_{l,i}(x) \leq h_{l,k}(x) \quad \forall i \in \text{pruned indices}.$$

Let $\tau(x) = h_{l,k}(x)$ be the retention threshold, then the pruned activation values $h_{l,i}(x) \leq \tau(x)$. The squared sum of the pruned values is:

$$\sum_{i \in \text{pruned indices}} h_{l,i}(x)^2 \leq \sum_{i=k+1}^{d_l} \tau(x)^2 = (d_l - k) \cdot \tau(x)^2.$$

Since $k = \lfloor r \cdot d_l \rfloor$, then $d_l - k \leq d_l(1 - r)$, therefore:

$$\sum_{i \in \text{pruned indices}} h_{l,i}(x)^2 \leq d_l(1 - r) \cdot \tau(x)^2.$$

According to the boundedness of gradients (Assumption 3), the derivative of the output with respect to the activation values is bounded. For any input $x$, we have:

$$\|\nabla F_n(w)\|_2 \leq G_f \quad \Rightarrow \quad \left\| \frac{\partial f_n}{\partial h_l} \right\|_2 \leq G_f.$$

Due to the chain rule in backpropagation, the dynamic range of the activation value $h_{l,i}(x)$ is limited by the gradient boundedness. Furthermore, assuming that network parameters are normalized (such as input normalization, weight initialization), the absolute value of the activation values does not exceed a certain constant $G_h$, i.e.:

$$h_{l,i}(x) \leq G_h \quad \forall i \in [d_l].$$

Therefore, the threshold $\tau(x) \leq G_h$. Substituting this into the previous equation, we get:

$$\sum_{i \in \text{pruned indices}} h_{l,i}(x)^2 \leq d_l(1 - r) \cdot G_h^2.$$

For input $x \sim D_n$, taking the expectation, we get:

$$\mathbb{E}_x \left[ \|h_l^{(r)}(x) - h_l(x)\|_2^2 \right] \leq \mathbb{E}_x \left[ d_l(1 - r) \cdot G_h^2 \right] = d_l G_h^2(1-r).$$

Define $C_l = d_l G_h^2$, then:

$$\mathbb{E}_x \left[ \|h_l^{(r)}(x) - h_l(x)\|_2^2 \right] \leq C_l(1 - r).$$

A.1.3. PROOF OF THEOREM 1

According to Assumption 1, each local loss function $F_n(w)$ is $L$-smooth. Therefore:

$$F(w^{t+1}) \leq F(w^t) + \langle \nabla F(w^t), w^{t+1} - w^t \rangle + \frac{L}{2} \|w^{t+1} - w^t\|_2^2.$$

Define the gradient of the full model as:

$$g_n(w^t) = \nabla F_n(w^t),$$

and the submodel gradient as:

$$\hat{g}_n(w^t) = \nabla \hat{f}_n(w^t; x_k).$$

According to Lemma 1, the gradient error satisfies:

$$\|g_n(w^t) - \hat{g}_n(w^t)\|_2 \leq G_h^2 H^2 \sum_{l,i} \|e_{l,i,k}\|_2.$$

According to Lemma 2, the expected squared norm of the activation error satisfies:

$$\mathbb{E}_x \left[ \|h_l^{(r)}(x) - h_l(x)\|_2^2 \right] \leq C_l(1 - r).$$

Therefore, the expected upper bound of the gradient error is:

$$\mathbb{E}_x \left[ \|g_n(w^t) - \hat{g}_n(w^t)\|_2^2 \right] \leq C_2(1 - r),$$

where $C_2 = G_h^4 H^4 \sum_l C_l$.

According to Assumption 2 (bounded gradient variance), the variance of the submodel gradients satisfies:

$$\mathbb{E}_x \left[ \|\hat{g}_n(w^t) - g_n(w^t)\|_2^2 \right] \leq \sigma^2.$$

Therefore, the variance of the aggregated gradient is:

$$\mathbb{E}_x \left[ \left\| \frac{1}{K} \sum_{k=1}^{K} \hat{g}_{n_k}(w^t) - \nabla F(w^t) \right\|_2^2 \right] \leq \frac{\sigma^2}{K}.$$

Substitute the submodel error and variance into the smoothness inequality:

$$F(w^{t+1}) \leq F(w^t) - \eta \langle \nabla F(w^t), \nabla F(w^t) \rangle + \frac{L\eta^2}{2} \left( \frac{\sigma^2}{K} + C_2(1 - r) \right).$$

After rearrangement:

$$F(w^{t+1}) \leq F(w^t) - \eta \|\nabla F(w^t)\|_2^2 + \frac{L\eta^2}{2} \left( \frac{\sigma^2}{K} + C_2(1 - r) \right).$$

Summing from $t = 1$ to $T$:

$$\sum_{t=1}^{T} \mathbb{E}[F(w^{t+1})] \leq \sum_{t=1}^{T} \mathbb{E}[F(w^t)] - \eta \sum_{t=1}^{T} \mathbb{E}[\|\nabla F(w^t)\|_2^2] + \frac{L\eta^2 T}{2} \left( \frac{\sigma^2}{K} + C_2(1 - r) \right).$$

Rearranging and using $F(w^{T+1}) \geq F(w^*)$:

$$\eta \sum_{t=1}^{T} \mathbb{E}[\|\nabla F(w^t)\|_2^2] \leq F(w^1) - F(w^*) + \frac{L\eta^2 T}{2} \left( \frac{\sigma^2}{K} + C_2(1 - r) \right).$$

After normalization:

$$\frac{1}{T} \sum_{t=1}^{T} \mathbb{E}[\|\nabla F(w^t)\|_2^2] \leq \frac{F(w^1) - F(w^*)}{\eta T} + \frac{L\eta}{2} \left( \frac{\sigma^2}{K} + C_2(1 - r) \right).$$

Let $C_1 = \frac{\sigma^2}{\mu} + \frac{LD^2}{\mu}$, $C_2 = G_h^2 H^2 \sum_l C_l$, and $C_3 = \frac{LG^2}{2}$. Define the compression factor $\mu = \frac{1}{K}$, the upper bound of parameter update step size $D$, and assume $F(w^1) - F(w^*) \leq \frac{\sigma^2}{\mu} + \frac{LD^2}{\mu}$, then:

$$\frac{1}{T} \sum_{t=1}^{T} \mathbb{E}[\|\nabla F(w^t)\|_2^2] \leq \frac{C_1}{\eta T} + C_2(1 - r) + \eta C_3,$$

where $\mu$ is the compression factor, $D$ is the upper bound of the parameter update step size.

## B. Discussion

### B.1. Communication Overhead

SpineFL employs a dual-strategy approach to optimize both communication efficiency and computational overhead. By training client-specific sub-models (pruned from a global backbone-dynamic layer split), the framework achieves significant computation reduction. Resource-limited clients completely avoid full-model training, while the activation-guided neuron selection mechanism introduces only minimal computational overhead.

For communication, SpineFL transmits only dynamic parameters (e.g., achieving a 70% reduction if dynamic layers occupy 30% of the model), unlike FedAvg's full-model updates.

### B.2. Privacy Protection

SpineFL enhances privacy preservation through its sub-model architecture. Each client trains only on a personalized subset of parameters (dynamic portion + client-specific backbone), which inherently limits exposure to the complete model structure. This design reduces the risk of model inversion attacks, as adversaries cannot reconstruct the full model from any single client's updates. Furthermore, the activation-guided selection mechanism operates on neuron-level statistics rather than raw data, adding an additional layer of privacy abstraction. During aggregation, parameter updates are masked through the `IsSel(·)` function, ensuring only locally relevant parameters are shared.

### B.3. Computational Overhead

SpineFL reduces computation primarily through training client-specific sub-models. The computational savings are

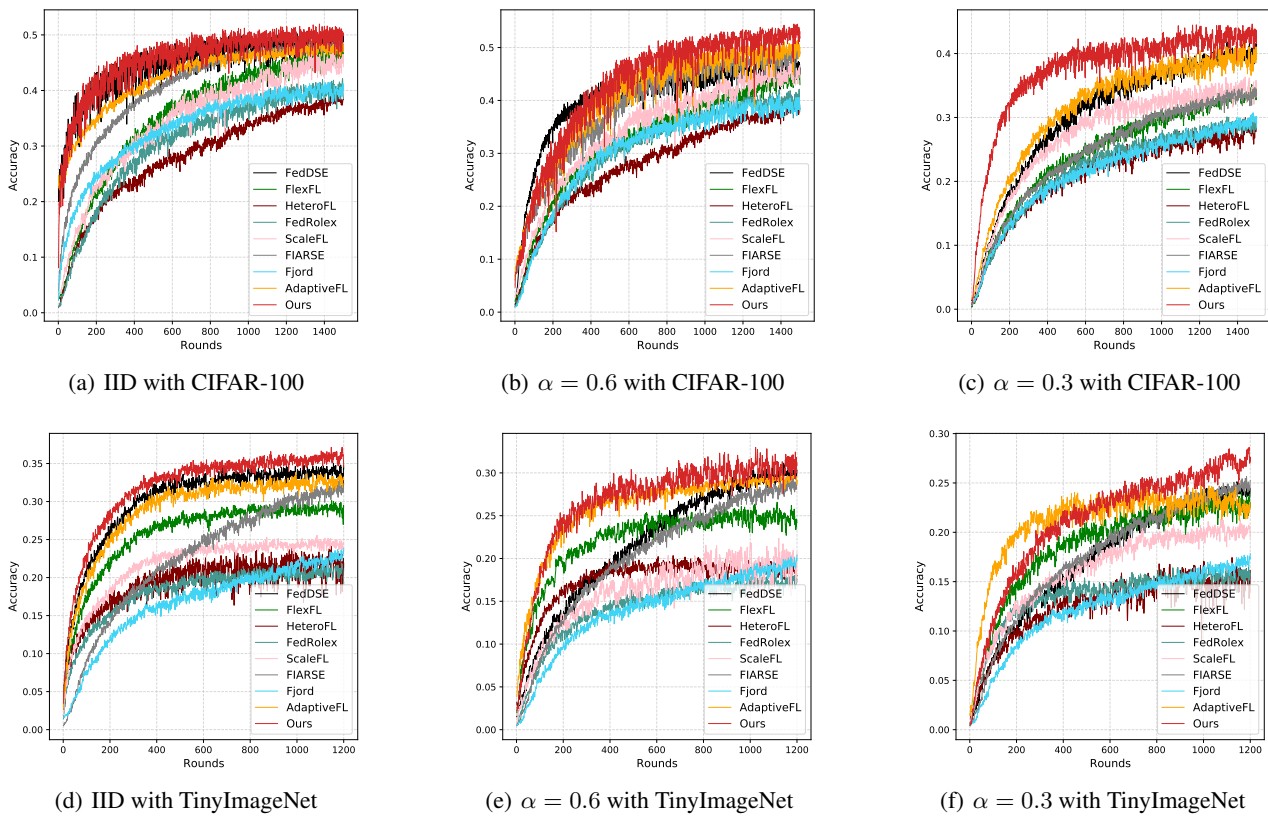

(a) IID with CIFAR-100     (b) $\alpha = 0.6$ with CIFAR-100     (c) $\alpha = 0.3$ with CIFAR-100

(d) IID with TinyImageNet     (e) $\alpha = 0.6$ with TinyImageNet     (f) $\alpha = 0.3$ with TinyImageNet

*Figure 6.* Learning curves of different FL methods using ResNet-34 on CIFAR-100 and TinyImageNet dataset

threefold:

Sub-model Size: Clients with limited resources (e.g., mobile devices) train significantly smaller sub-models (typically 30-70% smaller than full models).

Fixed Backbone: The backbone sharing mechanism (Algorithm 2) ensures shallow layer parameters remain fixed during local training, reducing backpropagation complexity.

Efficient Activation Calculation: The calculation of neuron importance scores ($s_{l,i} = \frac{1}{H \cdot W} \sum |A_d[l][i, j, k]|$) requires only a single forward pass per round, adding ¡5% overhead relative to local training time.

Optimized Server Operations: Server-side sub-model extraction is optimized through precomputed resource tables ($T_{res}, T_{act}$), limiting latency to ¡100ms per client even for models like ResNet-34.

### B.4. Limitations

The neuron selection mechanism prioritizes frequently activated neurons based on historical data, which may overlook rare or emerging patterns, reducing robustness in dynamic environments. At the same time, SpineFL assumes clients share the same global model structure, limiting its applicability to scenarios with inherently heterogeneous architectures

(e.g., clients using CNNs vs. Transformers). These will be an interesting topic for our future work.

## C. Additional Experimental Results

### C.1. Details of Baselines

We select the classic heterogeneous FL method HeteroFL together with four SOTA heterogeneous FL methods, i.e., FedRolex, FedDSE, ScaleFL, and FlexFL, as baselines.

- HeteroFL is a model-heterogeneous FL framework, which enables clients with varying computational capabilities to train submodels of different sizes (e.g., reduced width/depth) and resolves parameter dimension mismatches through hierarchical parameter alignment.

- FedRolex is a rolling submodule expansion-based FL method, which trains subsets of model layers or channels iteratively across clients, reduces local computation by partial parameter updates, and gradually expands global model capacity via shared parameter training.

- FedDSE is a dynamic sparse training framework, which dynamically selects critical subsets of model parameters (via gradient masking or importance scor-

ing) for local training and uploads sparse updates to reduce communication and computation costs while maintaining model performance.

- ScaleFL is a dynamic model scaling framework, which adaptively adjusts local model complexity (e.g., width/depth) based on client resources and aggregates heterogeneous submodels using layer-wise weighted averaging or parameter alignment strategies.

- FlexFL is an elastic federated learning framework, which allows clients to dynamically choose training configurations (e.g., epochs, batch sizes, or submodels) based on their resources and employs adaptive scheduling to optimize convergence under dynamic network conditions and Non-IID data.

## C.2. Learning Curves on CIFAR-100 and TinyImageNet dataset

Figure 6 presents the learning curves of SpineFL and all the baselines on CIFAR-100 and TinyImageNet datasets using ResNet-34. From the figure, we can observe that SpineFL can still achieve the best accuracy in all the cases. We can also observe that SpineFL can outperform all the baselines before they achieve their best accuracy.

## C.3. Impact of Neuron Selection Metrics

To further demonstrate the effectiveness of our proposed activation-guided selection mechanism, we compare the standard SpineFL with two alternative selection criteria: 1) Gradient-based Variation (which selects neurons based on gradient importance scores) 2) Deterministic Top-K Value (which deterministically selects the top-K neurons with the highest activation scores instead of using our probabilistic selection).

All variants are evaluated on the CIFAR-10 dataset under the IID setting using the ResNet-34 model. As shown in Table 3, SpineFL (Ours) achieves a test accuracy of 86.27%, outperforming the gradient-based method by 1.38% and the deterministic Top-K method by 1.04%. These results demonstrate that leveraging historical activation frequency alongside probabilistic selection strikes a better balance between feature importance and selection diversity, thereby validating the superiority of our approach.

*Table 3.* Test accuracy under different neuron selection

| Method | Gradient Variation | Top-K Value | Ours |
|--------|--------------------|-------------|------|
| **SpineFL** | 84.89 | 85.23 | 86.27 |

## C.4. Influence of temperature hyperparameter

All variants are evaluated on the CIFAR-10 dataset under the IID setting using the ResNet-34 model. For the temperature parameter T, we use the grid search between 0.5 to 2.0. A sensitivity analysis of the temperature parameter T was conducted, with results as follows: setting T=0.5 yields an accuracy of 85.18%, T=1.0 achieves the highest accuracy of 86.19%, and T=2.0 results in a reduced accuracy of 83.86%.

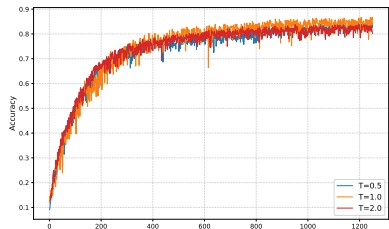

*Figure 7.* Influence of temperature hyperparameter

