# OpenReview forum: "Required Spine Optional Limbs: Heterogeneous Federated Learning via Backbone-sharing and Activation-guided Selection"
_ICML.cc/2026/Conference — ICML 2026 spotlight_

### Official Review · Reviewer_KCWH · 2026-03-10

**Soundness:** 3
**Presentation:** 3
**Significance:** 3
**Originality:** 3
**Overall Recommendation:** 5
**Confidence:** 3

**Summary:**

This paper studies federated learning under both model and data heterogeneity. The authors propose SpineFL, which divides the global model into a backbone shared by all clients and a dynamically selectable set of parameters. The proportion of the backbone is determined according to each client’s resource constraints, and parameters from the dynamic portion are then probabilistically selected based on neuron activation information. Finally, only the selected parameters are aggregated. Experimental results show that SpineFL achieves strong accuracy across multiple settings.

**Compliance With Llm Reviewing Policy:**

Affirmed.

**Final Justification:**

The authors’ response has addressed my concerns, and I have decided to keep my positive score.

**Key Questions For Authors:**

1. It is unclear whether the backbone parameters always participate in training or are frozen during local training.
2. Why is the aggregation not performed using the standard weighting by client sample size? This choice is particularly sensitive in non-IID settings.

**Limitations:**

yes

**Strengths And Weaknesses:**

Strengths: 1. First, the paper focuses on heterogeneous federated learning, which is itself an important direction and also a common real world scenario in AIoT and edge device federated learning.
2. The core improvement of SpineFL lies in a finer grained heuristic design for which parameters to share and which neurons to select. These two parts are intuitively easy to understand. The former emphasizes sharing common knowledge, while the latter emphasizes adapting to individual clients. The ablation study also shows that both parts play a role.
3. The experimental design is comprehensive. Results across different tasks, datasets, and experimental settings all demonstrate the effectiveness of the proposed method.

Weaknesses: 1. The related work review needs to be more up to date. More recent methods should be included, and some descriptions of prior work are confusing. For example, in the introduction, heterogeneous distillation usually requires a public dataset, whereas most prototype based methods may not require public data. This distinction should be made clear.
2. The definition of activation values and the way they are used are not sufficiently clear. It should be clearly explained whether the method uses the old activation values from the previous round or the newly computed activation values from the current round.
3. In the experimental section, the font size in the figures is too small, which makes them inconvenient to read.

---

> ### Author Rebuttal · Authors · 2026-03-30
>
> We kindly thank the reviewer for their careful review and insightful comments.
>
> **AW1:** Sorry for the confusion. We will revise the related work and introduction section to add more recent work. In addition, to demonstrate the effectiveness of our SpineFL, we compare SpineFL with two SOTA baselines, i.e., FedFree [R1] and FedConv [R2], on CIFAR-10 dataset using ResNet-34 model with $\alpha=0.3$ non-IID scenarios. The results as follows:
>
> | FedFree | FedConv | **SpineFL** |
> |:-:|:-:|:-:|
> | 75.24±1.62 | 74.39±1.19 | **78.30±1.46** |
>
> We can observe that SpineFL can still outperform both baselines.
>
> [R1] FedFree: Breaking Knowledge-sharing Barriers through Layer-wise Alignment in Heterogeneous Federated Learning. NeurIPS 2025
>
> [R2] FedConv: A Learning-on-Model Paradigm for Heterogeneous Federated Clients. ArXiv 2025
>
>
> **AW2:** Sorry for the confusion. SpineFL uses the old activation values from the previous round for submodel extraction. We will revise the manuscript to clarify that.
>
> **AW3:** Sorry for the confusion. We will revise the table and figures to make them easier to read.
>
> **AQ1:** We clarify that the backbone parameters always participate in local training and are not frozen. Specifically, the sub-model dispatched to each client is the union of the mandatory backbone portion and the dynamically selected neurons. During the local training phase (Step 4 in our workflow), the client performs standard backpropagation on the entire sub-model. The primary motivation for keeping the backbone trainable is to allow it to aggregate and learn generalized features from the diverse data distributions of all participating clients, serving as a stable bedrock for the global model.
>
> **AQ2:** We select simple averaging over sample-weighting mainly because i) the simple averaging does not require access to the information about the number of samples, and ii) using simple averaging can prevent the training towards specific clients with a large number of samples.

---

> > ### Author Rebuttal · Reviewer_KCWH · 2026-04-01
> >
> > Thanks to the authors for their response. I do not have any further queries or comments.

---

### Official Review · Reviewer_ks9j · 2026-03-12

**Soundness:** 4
**Presentation:** 3
**Significance:** 3
**Originality:** 3
**Overall Recommendation:** 5
**Confidence:** 4

**Summary:**

This paper proposes SpineFL, a sub-model extraction-based federated learning framework for heterogeneous and non-IID environments. By decomposing each model layer into a shared backbone and a dynamic portion, and using activation-guided pruning to generate client-specific sub-models, SpineFL mitigates parameter conflicts while preserving generalization. Experimental results demonstrate performance improvements over state-of-the-art heterogeneous FL methods.

**Compliance With Llm Reviewing Policy:**

Affirmed.

**Final Justification:**

After carefully reviewing the authors' rebuttal and re-evaluating the manuscript in light of their responses, I am pleased to confirm that my initial concerns have been fully and satisfactorily addressed. I now believe this work meets the standards for acceptance and will make a valuable contribution to the conference. I therefore strongly recommend accepting this paper.

**Key Questions For Authors:**

Please see weaknesses. If the author can address my concerns, I will increase the score.

**Limitations:**

yes

**Strengths And Weaknesses:**

Strengths:

1. The paper clearly identifies the limitations of existing sub-model extraction-based FL methods. The motivation for combining shared backbones with adaptive neuron selection is reasonable and intuitively grounded.

2. The method introduces a reasonable design that combines a globally shared backbone with a device-specific dynamic portion, effectively solving the non-IID data distribution.

3. Extensive experiments across multiple datasets, models, and heterogeneous data distributions demonstrate consistent and stable performance improvements over baselines.

Weakness:
1. Related work could be further expanded. The paper would benefit from a more comprehensive discussion of prior methods that utilize activation statistics, gradient-based importance, or other saliency measures for neuron or parameter selection, to better position the proposed activation-guided pruning strategy.

2. Sensitivity analysis could be more informative. While Section 5.4 discusses the impact of the hyperparameter T, the presentation could be improved by including corresponding figures, such as learning curves or accuracy trends under different T values. This would provide clearer insights into the training dynamics and convergence behavior.

3. Evaluation under more extreme non-IID settings could be explored. The experiments mainly consider moderate levels of data heterogeneity. It would be helpful to further evaluate the method under more extreme non-IID scenarios ($\alpha$=0.1), to better understand its robustness in highly skewed settings.

4. Typos. Line 392 and Line 410 “six methods” -> “nine methods”?

---

> ### Author Rebuttal · Authors · 2026-03-30
>
> We kindly thank the reviewer for their careful review and insightful comments.
>
> **AW1:** Thanks for your suggestion. To demonstrate the effectiveness of the activation-guided selection mechanism, we compare the original SpineFL with the selection mechanism using the metric of gradient or Top-K value on the CIFAR-10 dataset under IID settings using the ResNet-34 model. The results are as follows:
>
> | Method | Acc |
> |:-:|:-:|
> | Gradient Variation | 84.89 |
> | Top-K Value | 85.23 |
> | **Ours** | **86.27** |
>
> We can observe that the activation-guided selection mechanism outperforms that using gradient or Top-K value. In addition, we will conduct in-depth research and incorporate discussions on saliency measures, including gradient-based methods and neuron evaluation methods commonly used in traditional pruning strategies.
>
> **AW2:** Sorry for the confusion. Due to the space limitation, we only present the impact of the hyperparameter T through text and tables in Section 5.4. In the final version, we will supplement with learning curves and an accuracy trend figure to showcase the convergence speed and final performance of the model under different T values.
>
> **AW3:** Thanks for your suggestion. To demonstrate the performance of SpineFL on extreme non-IID scenarios, we conduct experiments on CIFAR-10 dataset with $\alpha=0.1$ using ResNet-34 model. The results as follows:
>
> | HeteroFL | FedRolex | FedDSE | ScaleFL | FlexFL | Fjord | AdaptiveFL | FIARSE | **SpineFL** |
> |:-:|:-:|:-:|:-:|:-:|:-:|:-:|:-:|:-:|
> |48.26±2.58|50.17±2.64|61.82±2.15|60.35±1.97|65.47±2.03|48.14±1.89|62.83±1.77|61.29±1.92|**68.59±1.76**|
>
> We can observe that SpineFL can still outperform all the baselines.
>
> **AW4:** Sorry for the confusion. We will revise the manuscript to fix all the typos.

---

> > ### Author Rebuttal · Reviewer_ks9j · 2026-04-02
> >
> > Thank you for the detailed rebuttal. My main concerns have been satisfactorily addressed. I will keep my positive score.

---

### Official Review · Reviewer_aP48 · 2026-03-12

**Soundness:** 3
**Presentation:** 3
**Significance:** 3
**Originality:** 3
**Overall Recommendation:** 5
**Confidence:** 4

**Summary:**

The paper presents SpineFL, a novel federated learning framework designed to address the challenges posed by heterogeneous devices and non-IID data. It introduces a backbone-sharing mechanism that decomposes each global model layer into a mandatory shared portion for generalization and a dynamic portion for personalization. To generate sub-models, SpineFL employs an activation-guided selection strategy that probabilistically chooses neurons from the dynamic portion based on their historical activation frequency, which mitigates parameter conflicts while maintaining selection diversity. Experimental results demonstrate that SpineFL surpasses all the eight baslines.

**Compliance With Llm Reviewing Policy:**

Affirmed.

**Final Justification:**

Please see the Rebuttal Acknowledgement.

**Key Questions For Authors:**

Please clarify the weaknesses.

**Limitations:**

yes

**Strengths And Weaknesses:**

Strength:

1. The motivation of this paper is clear and the proposed method is novel and reasonable.

2. This paper is well-written and easy to follow.

3. The author present the convergence analysis.

4. The author conduct comprehensive experiments across multiple datasets and the experiments demonstrate the effectiveness of the proposed method.

Weakness:

1. The effectiveness of the activation-guided selection mechanism could be further clarified. Could the authors further clarify the effectiveness of the activation-guided selection mechanism? Specifically, is activation-based importance superior to other commonly used criteria, such as gradient-based importance or deterministic top-K selection based on activation scores?

2.  Minor: Although the author compared with sufficient baselines, none of them are published in 2025. I suggest the author disscuss more works published in 2025.

---

> ### Author Rebuttal · Authors · 2026-03-30
>
> We kindly thank the reviewer for their careful review and insightful comments.
>
> **AW1:** Sorry for the confusion. To demonstrate the effectiveness of the activation-guided selection mechanism, we compare the original SpineFL with the selection mechanism using the metric of gradient or Top-K value on the CIFAR-10 dataset under IID settings using the ResNet-34 model. The results are as follows:
>
> | Method | Acc |
> |:-:|:-:|
> | Gradient Variation | 84.89 |
> | Top-K Value | 85.23 |
> | **Ours** | **86.27** |
>
> We can observe that the activation-guided selection mechanism outperforms that using gradient or Top-K value.
>
> **AW2:** Thanks for your suggestion. To demonstrate the performance of SpineFL, we compare SpineFL with two baselines, i.e., FedFree [R1] and FedConv [R2], on CIFAR-10 dataset using ResNet-34 model with $\alpha=0.3$ non-IID scenarios. The results as follows:
>
> | FedFree | FedConv | **SpineFL** |
> |:-:|:-:|:-:|
> | 75.24±1.62 | 74.39±1.19 | **78.30±1.46** |
>
> We can observe that SpineFL can still outperform both baselines.
>
> [R1] FedFree: Breaking Knowledge-sharing Barriers through Layer-wise Alignment in Heterogeneous Federated Learning. NeurIPS 2025
>
> [R2] FedConv: A Learning-on-Model Paradigm for Heterogeneous Federated Clients. ArXiv 2025

---

> > ### Author Rebuttal · Reviewer_aP48 · 2026-04-02
> >
> > I thank the authors for their effort and detailed rebuttal. Most of my concerns have been satisfactorily addressed, and I will maintain my initial positive score.

---

### Official Review · Reviewer_NyJc · 2026-03-12

**Soundness:** 3
**Presentation:** 3
**Significance:** 3
**Originality:** 2
**Overall Recommendation:** 4
**Confidence:** 4

**Summary:**

The paper introduces SpineFL, a heterogeneous federated learning framework that splits the global model into a shared backbone, trained by all clients to improve generalization, and a dynamic portion, from which neurons are selected using an activation‑guided strategy to match each device’s resource constraints. Experiments on several datasets and architectures show that SpineFL outperforms existing sub‑model extraction baselines by reducing parameter conflicts and improving robustness under moderate heterogeneity.

**Compliance With Llm Reviewing Policy:**

Affirmed.

**Final Justification:**

Authors addressed my concerns and some ambiguities I had.

**Key Questions For Authors:**

- How are device resource capacities actually measured or estimated in practice?
- Why do the experiments use random resource percentages, and how do these map to real devices?
- How sensitive is SpineFL to errors or noise in resource estimation?
- How does SpineFL handle dynamic or fluctuating device resources during training?
- Can the authors test SpineFL under more extreme non‑IID conditions (e.g., single‑label clients or severe imbalance)?

**Limitations:**

The paper briefly discusses one limitation in the appendix. The paper could add briefly in the conclusion some limitations, and a potential impact on fairness in the impact statement:
1) Adding details about where SpineFL may fail, particularly under harsh non-IID conditions or hardware deployment related issues which were not included in the experiments (thermal throttling, battery drain…). Could also be highlighted in possible experiments.
2) Potential fairness or societal consequences related to model quality differences across devices with unequal capabilities and with minority represented data. It could also be interesting to mention Mitigation strategies (e.g., fairness-aware aggregation) to address these risks.

**Strengths And Weaknesses:**

*Soundness:*
 The paper presents a technically coherent method with supporting theory and experiments; however, the experimental evaluation is limited to scenarios likely favorable to the approach, and the authors do not test or discuss potential failure cases.
*Presentation:*
 Overall clear and well-organized and the  algorithms and components are clearly described, but limitations could be better communicated.
*Significance:*
 Addresses a current problem in heterogeneous FL but offers an incremental improvement.
*Originality:*
 Provides a novel combination of backbone-sharing and activation-guided selection that extends prior work, though the components themselves are adaptations of known techniques. I am not very familiar with the literature on this specific area, but the proposed technique seems very similar to the structured updates based on masks proposed in the widely cited (Konečný et al 2016).
Konečný, J., McMahan, H. B., Yu, F. X., Richtárik, P., Suresh, A. T., & Bacon, D. (2016). Federated learning: Strategies for improving communication efficiency.

---

> ### Author Rebuttal · Authors · 2026-03-30
>
> We are grateful to the reviewer for their careful review and insightful comments.
>
> **Answer for Originality:** Sorry for the confusion. We need to clarify that our SpineFL is significantly different than structured update methods [Konečný et al. 2016].
>
> - First, the goals of the two works are different. SpineFL aims to address the device heterogeneity issue, where clients train structurally different subnetworks derived from a shared backbone to match their resource constraints. The structured update method aims to reduce communication overhead, which assumes the local training is still using the same full model rather than a submodel.
>
> - Second, the methodologies of the two works are different. SpineFL leverages neuron responses during forward propagation to estimate their contribution, enabling data-driven and semantics-aware structural selection, rather than relying on random or magnitude to select parameters. In addition, SpineFL adopts the resource-aware dynamic subnetwork construction to map device constraints (e.g., computation or memory budgets) to subnetwork configurations, producing models that are structurally different yet functionally aligned for local training on heterogeneous devices rather than fixing specific parameters on local training.
>
> Therefore, while both approaches involve partial parameter selection, SpineFL is significantly different from structured update methods.
>
> **AQ1:** In reality, the available resources on a client can be estimated based on the model size uploaded by the client, or the client can proactively upload information about available resources.
> Please note that SpineFL supports the clients in pruning their received model.
>
> **AQ2:** Following the various existing related works, such as HeteroFL, AdaptiveFL, and FlexFL, we adopt the random resource percentages to simulate the device resource rather than using the parameters of specific real devices.
>
> **AQ3:** Sorry for the confusion. If the available resource of the client cannot support the training of the received sub-model, the client can prune the model using the activation-guided neuron selection strategy in Sec. 3.2.2 based on its stored activation information.
>
> **AQ4:** Sorry for the confusion. Since sub-model extraction is performed at each training round based on the latest resource table, SpineFL can update the resource table in real time when resources change. In addition, as mentioned in AQ3, clients can prune the received model based on their available resources. Therefore, SpineFL can deal with the scenarios of fluctuating resources.
>
> **AQ5:** Thanks for your suggestion. To demonstrate the performance of SpineFL on extreme non-IID scenarios, we conduct experiments on CIFAR-10 dataset with $\alpha=0.1$ using ResNet-34 model. The results as follows:
>
> | HeteroFL | FedRolex | FedDSE | ScaleFL | FlexFL | Fjord | AdaptiveFL | FIARSE | **SpineFL** |
> |:-:|:-:|:-:|:-:|:-:|:-:|:-:|:-:|:-:|
> |48.26±2.58|50.17±2.64|61.82±2.15|60.35±1.97|65.47±2.03|48.14±1.89|62.83±1.77|61.29±1.92|**68.59±1.76**|
>
> We can observe that SpineFL can still outperform all the baselines.

---

> > ### Author Rebuttal · Reviewer_NyJc · 2026-04-03
> >
> > Authors have addressed my comments and clarified some ambiguities i had.

---

### Decision · Program_Chairs · 2026-04-30

**Decision:**

Accept (spotlight)

**Comment:**

All reviewers agree with the significant novelty of the paper and achieve a consistent decision. The responses of the author also address some minor flaws.